# STBEAT: Software Update on Trusted Environment Based on ARM TrustZone

**Qi-Xian Huang** [1] , **Min-Yi Chiu** [1] , **Chi-Shen Yeh** [2] **and Hung-Min Sun** [3,*]

1 Institute of Information Systems and Applications, National Tsing Hua University, Hsinchu 300, Taiwan
2 Institute of Information Security, National Tsing Hua University, Hsinchu 300, Taiwan
3 Department of Computer Science, National Tsing Hua University, Hsinchu 300, Taiwan
* Correspondence: hmsun@cs.nthu.edu.tw

**Abstract:** In recent years, since edge computing has become more and more popular, its security issues have become apparent and have received unprecedented attention. Thus, the current research concentrates on security not only regarding devices such as PCs, smartphones, tablets, and IoTs, but also the automobile industry. However, since attack vectors have become more sophisticated than ever, we cannot just protect the zone above the system software layer in a certain operating system, such as Linux, for example. In addition, the challenges in IoT devices, such as power consumption, performance efficiency, and authentication management, still need to be solved. Since most IoT devices are controlled remotely, the security regarding system maintenance and upgrades has become a big issue. Therefore, a mechanism that can maintain IoT devices within a trusted environment based on localhost or over-the-air (OTA) will be a viable solution. We propose a mechanism called STBEAT, integrating an open-source project with ARM TrustZone to solve the challenges of upgrading the IoT system and updating system files more safely. This paper focuses on the ARMv7 architecture and utilizes the security stack from TrustZone to OP-TEE under the STM32 board package, and finally obtains the security key from the trusted application, which is used to conduct the cryptographic operations and then install the newer image on the MMC interface. To sum up, we propose a novel software update strategy and integrated ARM TrustZone security extension to beef up the embedded ecosystem.

**Keywords:** ARM TrustZone security; access control; data security; embedded software; embedded system; STM32

## 1. Introduction

Due to the growing market for the internet of things, most internet of things devices use ARM architecture to implement their products. ARM is a family of reduced instruction set computer (RISC) architecture, which has several advantages as shown below:

1. The instruction length is fixed so that the difficulty to design the instruction decoder can be reduced;
2. It is a load/store machine, which means all data-processing operations can only be operated on the registers, which could optimize the latency;
3. Small code size with a highly optimized set of instructions, such as a combination between arithmetic and logical operations on shift instructions.

There are several generations of ARM design. The architecture of each generation comes with subtly different profiles: (1) "Application profile"—it supports the virtual memory system architecture based on an MMU, and both ARM and Thumb instruction sets as well; (2) "Real-time profile"—it serves a protected memory system architecture based on an MPU; (3) and "Microcontroller profile"—this model is designed for fast interrupt processing and easy integration into an FPGA for processors and is suitable for low power applications [1].

This research focuses on ARM TrustZone technology, which has been proposed since ARMv6 architecture. TrustZone is a security extension of ARM System-On-Chip (SoC) covering the processors, memory, and peripherals, dividing them into the normal world and the secure world [2]. As can be expected, the normal world cannot directly read from or write to the secure world or perform any direct access operations. This feature has been defined as the trusted execution environment (TEE) by the GlobalPlatform which has spent a lot of time and resources standardizing the TEE internal API. This organization emphasized "the TEE is a secure area of the main processor of a device and must offer isolated safe execution of authorized security software" [3].

Basically, STBEAT is based on the integration SWUpdate [4] and OP-TEE [5,6] projects to create a trusted update environment and relies on the original mechanism to avoid corruption of the update procedure. In the past, upgrading or patching a system was challenging. Once the update process fails, the system will corrupt. IoT devices have low computing power to complete complex mechanisms, but we want to make software upgrades easy and secure to deploy.

The MPU of STM32 [7,8] is based on the ARM Cortex-A profile, which uses the ARM TrustZone architecture to isolate resources. As shown in Figure 1, here we take ARMv8-A as an example. It defines several exception levels as follows:

1. EL0 is the lowest execution level, allowing applications to make unprivileged calls.
2. EL1 on the normal side is the execution level of the normal operating system.
3. EL1 on the secure side is the exception level for secure monitor execution.
4. EL2 is the hypervisor layer and is only used for the non-secure world.

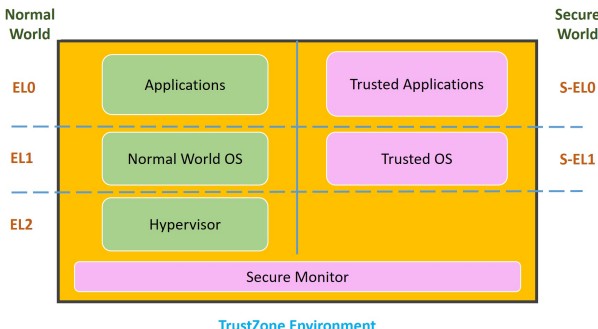

**Figure 1.** Definition of the ARMv8-A exception level [9].

The Linux kernel and its application framework belong to the normal world, on the other hand, the secure monitor with minimal services or trusted operating system belongs to the secure world.

Thanks to the TrustZone [10–12] support of the STM32MP157 microprocessor, our system protection solution is all encompassing and not limited to the CPU context. The architecture provides the bus and peripheral infrastructure to ensure that the secure world uses a fully secure pipeline to control secure peripherals. For example, internal or external peripherals can be used by the secure world to support cryptographic operations.

The TrustZone divides the environment into a secure world and a non-secure world. The secure world guarantees code and data integrity with hardware support to isolate the context of the CPU. The protected resources can be DDR locations, SoC peripheral interfaces, or SoC internal resources. Basically, STBEAT is based on the integration SWUpdate and OP-TEE projects to create a trusted update environment and rely on the original mechanism to avoid the update corruption. In the past, upgrading or patching a system used to be a challenge. Once the update process fails, the system will corrupt, but IoT devices have the low computing power to complete complex mechanisms, which we want to make software upgrades easy and secure to deploy.

### 1.1. SWUpdate Project

The SWUpdate project is a mature OTA updater for embedded Linux devices and is also a completely open-source project. The code of SWUpdate is released under open-source licenses and is integrated with other free and open source software (FOSS) [13] projects. The SWUpdate has many benefits, such as the following:

1.  It is careful about using resources. It has a small footprint and low usage of resources and can be integrated into devices with limited memory and flash storage. Its zero-copy option allows the installation of new software without the need for temporary copies and access to resources.
2.  The SWUpdate is developed with security in mind to prevent unauthorized software from running on your device.
3.  SWUpdate is highly customizable and flexible so that you do not need to change your project for SWUpdate. You can adapt it to some specific update requirements.
4.  It is easy to use and has many use cases. You can update from a USB stick, or need a management system to deploy.

The main functional details of SWUpdate are as follows:

1.  The SWUpdate supports signing with RSA keys and with certificates using your own PKI infrastructure to validate authorized and trusted updates with signed and verified update packages.
2.  The rollback mechanism detects whether the installed software is working properly and rolls back to the previously installed version in case of failure.
3.  SWUpdate supports zero-copy, which can be installed directly on the storage without creating temporary files.
4.  SWUpdate supports two ways to update your system. One is an offline update, such as USB, SD, and various other local interfaces. The other is the OTA update, which has an integrated webserver to upgrade the device.

The SWUpdate has been widely used to provide a secure and reliable way to update products, and due to its high flexibility, we do not need any special requirements and it can be integrated into any embedded system project. That is the reason we chose this open source project to improve security by supporting TrustZone.

### 1.2. Contributions

This paper utilizes the TrustZone extensions and integrates OP-TEE designed primarily to rely on ARM Trust- Zone technology into STBEAT to develop a better and more secure way to update software [14]. The SWUpdate is an open-source project aimed to provide a full-scale solution to upgrade embedded systems and their files. Nevertheless, SWUpdate runs on Linux under the normal world of the ARM core, and we made practical improvements to SWUpdate to isolate the crypto operations and added them to trusted applications.

When SWUpdate is triggered, it will create a channel between the normal and secure sides, and then SWUpdate communicates with the trusted application and uses the predefined commands to execute further installation. More details are provided in [9,15,16]. To sum up, our contributions are as follows:

1.  We implement the prototype of the software update mechanism through ARM Trust-Zone, and also modify the SWUpdate client of the open source software update project to meet our requirements.
2.  We regard security as the default option of enterprise products.
3.  We integrate OP-TEE and SWUpdate to make the embedded ecosystem more secure because we directly modified the upstream project, so developers can use the project without modification.

We summarize the concept of TEE, which is based on ARM TrustZone technology, and TrustZone architecture, which is the system design solution from ARM. In addition, there are several projects of TEE, for example, Linux is based on OP-TEE, Android is based

on Trusty, a specific vendor such as Samsung is based on TZ-RKP, and Qualcomm is based on QSEE, to name a few. Currently, the system software threat is no longer merely hijacking the root privileges because the critical data has been placed in the TEE environment to separate resources and provide secure services including user authentication, usage of secure resources, and trusted isolation and processing.

Due to the above-mentioned restricted access mechanisms, we can put critical calculations into the secure world to protect data effectively, even if the normal world has the highest privilege to take control. If we want to obtain the kinds of output such as calculation results or private data from the secure world while in the normal world, one way to achieve our goal is to trigger the TEE driver to make a secure monitor call and switch the control to the secure world. We built the software update application into the secure world, making every transaction of the update procedure secure. We improved security and made them the default option for commercial products.

In the following sections of this paper, we will introduce the required background knowledge of this paper. Section 2 introduces the SDK from STM32, TEE implementation, SWUpdate project and reviews the system architecture and related works. Sections 3 and 4 present experiments to demonstrate the integration, and evaluate whether the result is successful and meaningful. Finally, Sections 5 and 6 summarize the fundamental framework contribution and give directions for future work.

## 2. Related Works

### 2.1. STM32MP15 Microprocessor

As the specification of STM32MP15 [17] microprocessors, they are based on the Arm Cortex-A7 [13] dual core. Furthermore, there are many product lines to meet the different requirements of customers, such as the following:

1. STM32MP151: Single Cortex-A7 with Cortex-M4, without GPU.
2. STM32MP153: Dual Cortex-A7 with Cortex-M4, without GPU.
3. STM32MP157: Dual Cortex-A7 with Cortex-M4, with GPU.

Additionally, different types have different security functions and CPU clock frequencies.

1. STM32MP15xA: only have basic security functions, with clock rate of 650 MHz.
2. STM32MP15xC: have secure boot and cryptography module, with clock rate of 650 MHz.
3. STM32MP15xD: only have basic security functions, with clock rate of 800 MHz.
4. STM32MP15xF: have secure boot and cryptography module, with clock rate 800 MHz.

All in all, as shown in Figure 2, the STM32MP15F block diagram describes that it supports more security extensions. Including TrustZone, we focused on, in this paper, the AES/DES hardware module, secure ROM and RAM, and peripherals.

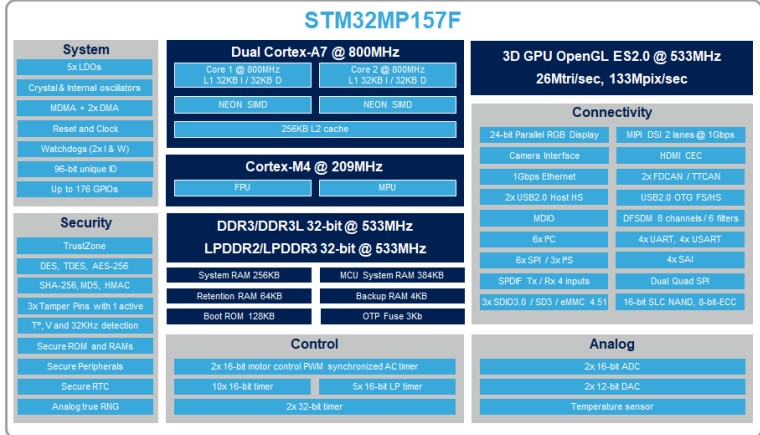

**Figure 2.** STM32MP157F block diagram.

### 2.2. Scratch Block Mechanism to Prevent System Damage

Since various update strategies have been proposed, the relevant distinguishing points among them are the encryption key and the update methods. This approach develops a reliable mechanism to divide the flash memory into the current boot area and the storage area, which is ready to be updated [18]. The agent on the normal world side is responsible for communicating with the remote server and downloading the new software and then installing it in the local storage. After that, the client sends the notification to the device for a reboot. During the update process, the trusted application will check the metadata of the software and verify the hash value or signature of the entire image.

Further, when beginning the new software update, there are three regions allocated for the new image block, old image block, and scratch block. The new image block is the place where we want to flash with a new software package; the old image block is the system currently running; the scratch block is the temporary buffer employed to swap in and swap out. This mechanism is designed to prevent system corruption caused by power loss or software update errors during updates.

We begin by copying the new image block to the scratch area first, then copying the old image block to the new image block, and finally copying the scratch block to the old image block; we just keep cycling through these steps. After the exchange of each block, the relevant information is recorded in a reserved memory area.

If there is a power failure or an aborted update during the update process, the boot-loader will obtain the recorded information from the previous update and continue the update process. Therefore, this mechanism can ensure the integrity of software updates.

Moreover, there are many encryption and decryption processing operations in a secure system, including encrypted data storage or secure connections to remote servers. A secure and isolated system offers the most significant improvement in key management, and we can place critical keys into the secure world. At this moment, the non-secure world has to request the encryption or decryption operation and the specific encryption or decryption handlers to perform it.

### 2.3. Secure Transmission and Multiple Encryption Support

Since the solution mentioned above is not comprehensive enough, this research proposes a novel and more secure update process to compensate for the shortcomings [19]. When the host application starts to update the system, it will securely download firmware using an HTTPS connection from a remote server. After downloading system images, the host application will divide the firmware into several blocks and send each block to a trusted application to calculate the SHA256 checksum.

This approach also defines the signature generated by the computational checksum using firmware and the RSA public key. Hence, the trusted application compares the checksum with the signature, which is passed by the host application. If all verification processes are passed, the trusted application will write the firmware to the MMC interface or NAND flash memory through their driver in the secure world.

### 2.4. Boot Chain Overview [20]

As shown in Figure 3, the first startup program is ROM code, also known as the ROM stage. The ROM code is a piece of software that is stored in the read-only memory (ROM). The ROM code is the first executed by the processor, and it will select the boot device as the first-stage boot loader (FSBL) to load into embedded RAM. In addition, it will perform the basic clock tree initialization and FSBL loading from the boot device and FSBL launch in the ROM stage.

The next stage is FSBL, which will complete the initialization of the clock tree and the external RAM controller. After initialization, the FSBL will load the second-stage boot loader (SSBL) into the external RAM and jump to it.

The next stage is SSBL, which runs in wide RAM. It can support complicated features, such as USB, Ethernet, and display. The U-Boot is commonly used for the Linux bootloader in this stage.

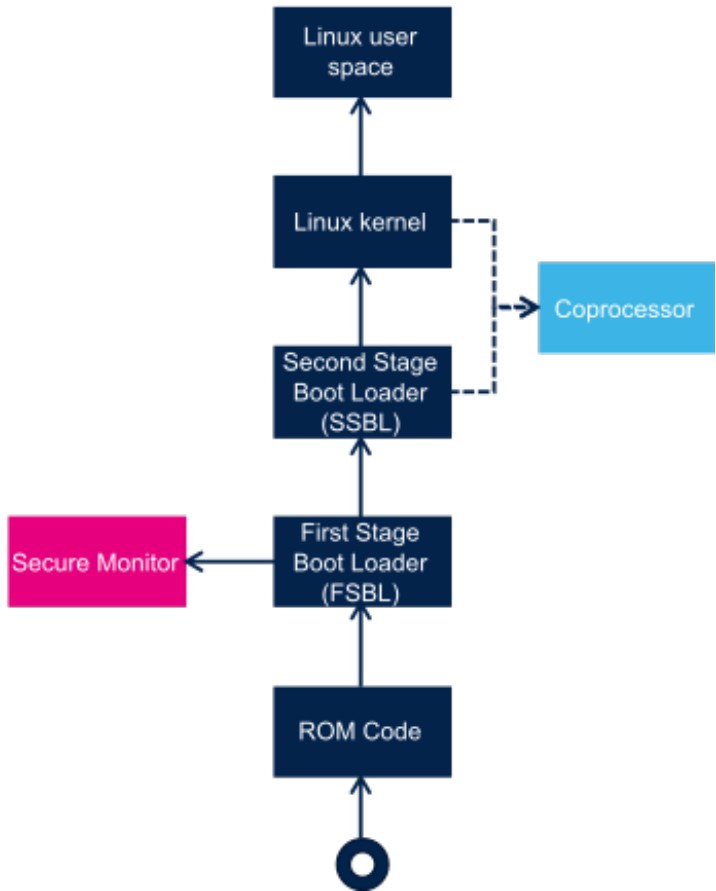

**Figure 3.** The boot chain of STM32MP15.

*2.5. Secure Boot [21]*

The STM32 MPU also supports the secure boot mechanism to ensure the integrity of the platform at runtime. The STM32 trusted boot chain is outlined below:

1. Configuration of execution rules of the platform, the rules are the essential elements for a safe execution on the platform.
2. The integrity and authentication, which uses hash algorithms and asymmetric cryptography algorithms to verify boot software components.

The OpenSTLinux is a platform for the STM32 MPU that uses TF-A for trusted boot and peripheral access control.

As we know, there are two solutions to provide services in terms of security. One of the runtime services is TF-A SP_MIN, which is a minimal secure service provider, provided by TF-A, including PSCI controls, SCMI resources, and other services for power states transition or platform facilities. Another secure service provider is the OP-TEE operating system, recommended by STMicroelectronics. It is an open-source TEE solution that can run core secure services and trusted applications, respectively. The trusted applications are used as secure components, exposing generic services to the non-secure side, such as random number generation or other cryptographic operations.

The mandatory step to ensure a secure boot is to load the TF-A BL2 firmware into RAM via ROM, so BL2 firmware needs to be encapsulated in a binary file that starts with an STM32 header that is able to authenticate and boot the firmware.

The further boot process will go to BL32, which can be either SP-MIN or OP-TEE, and the next stage is BL33, which boots the normal operating system via U-Boot.

In summary, as shown in Figure 4, the boot chain of STM32MP15 uses TF-A as the first-stage bootloader, and it uses U-Boot as the second-stage bootloader. In addition, we can enable or disable the secure boot mechanism so that we can run a secure variant on any STM32MP15 device to accommodate.

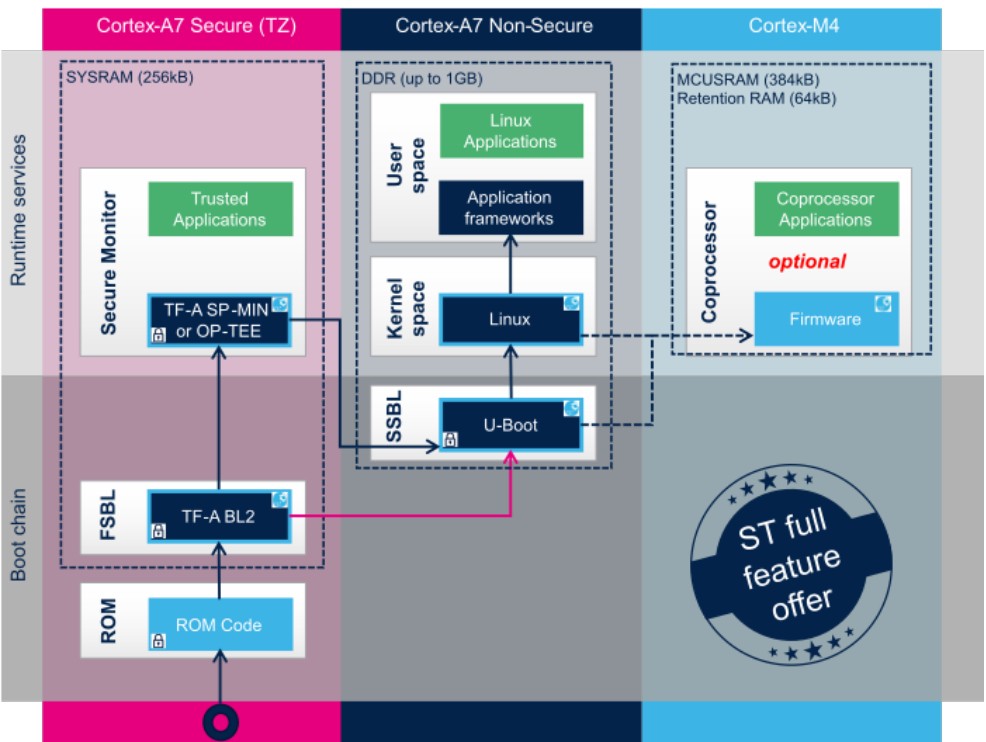

**Figure 4.** The components for secure boot of STM32MP15.

## 3. Methodology

### 3.1. Architecture

As the architecture overview in Figure 5 shows, we use a typical operating system such as Linux in the normal world. Moreover, we use the OP-TEE operating system as a server to serve the normal world. We propose this strategy and conducted experiments on the STM32MP1 family of platforms with two ARM cores, Cortex A7 and Cortex M4. Our experiments do not use the Cortex M4 subsystem instead using Cortex A7, which is a 32-bit ARMv7-A architecture microprocessor with TrustZone support. We use the arm trusted firmware (TF-A) to plan resource allocation and assign resource access control policies to boot up the platform. After booting TF-A, the secure OS OP-TEE will be loaded into memory and then controlled to U-Boot, which is used as the bootloader for the normal world operating system.

Once all startup processes are complete, the SWUpdate client is in standby mode as a daemon process and waits for update events. If the client application is called by an event, it will trigger a system call to trap into kernel mode. After the soft interrupt is processed, the next step is to find out the tee driver to make a secure monitor call instruction to change the world state.

All in all, the SWUpdate client and trusted application are located in the normal world and the secure world, respectively. The update process relies on the above two components, the TEE driver within the Linux kernel and the OP-TEE operating system, which is responsible for handling requests from the TEE driver and trusted applications.

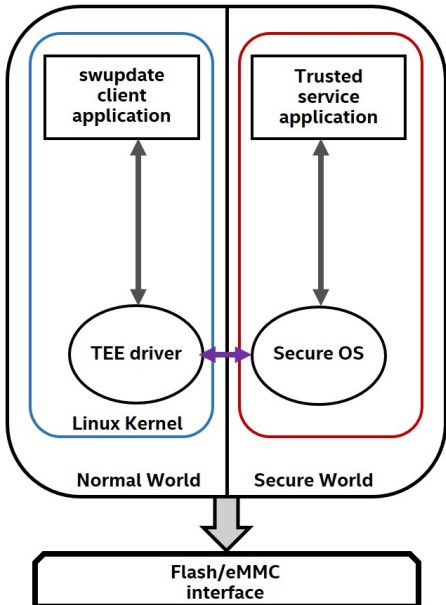

**Figure 5.** Software update process architecture, which has the SWUpdate client inside the normal world and trusted application inside the secure world.

### 3.2. Tee-Supplicant

The tee-supplicant is one handler of trusted application requests, as shown in Figure 6. TA can send the remote procedure call to the tee-supplicant, which is constructed by an infinite loop to process requests from TA. As shown in Figure 5, we use the method to load TA dynamically; thus, when the SWUpdate initializes the context, TA will be loaded with universal unique identification (UUID) passed by the SWUpdate client and put into the secure world by the tee-supplicant. When TA is initialized, a channel is established between the normal and the secure world to exchange information.

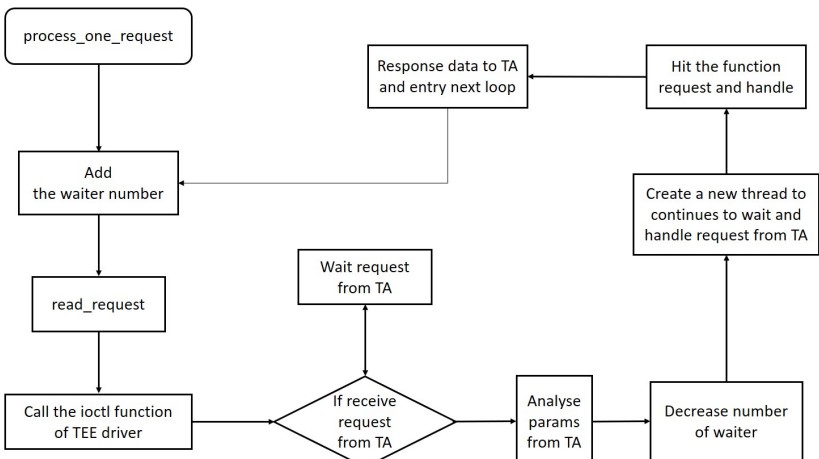

**Figure 6.** The communication process of tee-supplicant.

### 3.3. SWUpdate

SWUpdate is an open-source project based on the GPLv2 license and a Linux-based update agent, which supports multiple update strategies on both local and over-the-air (OTA). Moreover, SWUpdate has many features, such as the ability to update the roots and kernel of the device, the ability to install the system files into embedded media (eMMC/SD/NAND/NOR), and some of the embedded media requiring the memory technology device (MTD) library for bridging. This paper relies on the single copy mechanism

to update the system and files, but we still plan to provide a comprehensive solution by the dual-copy mechanism to reduce the impact on system corruption or critical file leakage.

### 3.4. OP-TEE Architecture

The OP-TEE project includes many secure and non-secure components to support trusted applications. As shown in Figure 7, the main components of OP-TEE are the OP-TEE core and shared libraries of trusted applications on the secure side, and the client API library, which is constructed by the OP-TEE supplicant and the OP-TEE Linux kernel driver on the non-secure side.

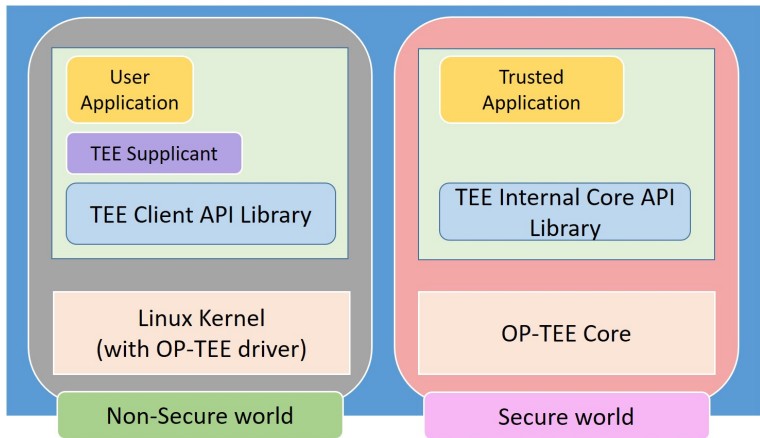

**Figure 7.** The OP-TEE architecture of STM32MP15 [15].

As mentioned above, the OP-TEE core executes in secure privileged mode, while trusted applications are executed in secure user mode. OP-TEE can load trusted applications stored in the file system of the Linux operating system or the OP-TEE core boot image. Trusted libraries include the TEE internal core API libraries provided by OP-TEE for trusted application development, and OP-TEE also supports the loading of static and dynamic libraries in the TEE.

The OP-TEE core can use non-secure userland supplicant, which can be invoked through the OP-TEE Linux kernel driver. A scenario for this service is that we need to access a non-volatile device that is controlled on the non-secure side, so we need the suppliant to handle this request from a trusted application.

OP-TEE is initialized and ready to serve when the Linux kernel is booted. As shown in Figure 8, the TEE driver in the Linux kernel is a generic API, which is exposed from the "libtee" library and is the interface among the SWUpdate client, the trusted OS, and the tee-supplicant.

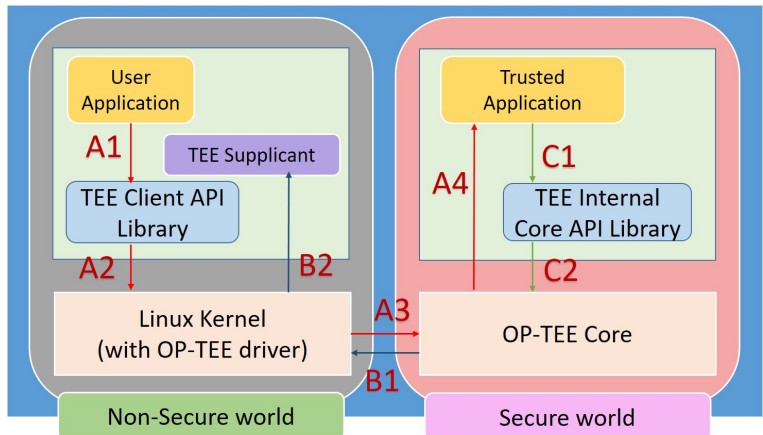

**Figure 8.** The location of the TEE drive and its control flow [15].

There are several scenarios that meet our proposal:

1.  For a non-secure application, services are invoked from a trusted application. The non-secure application first calls the TEE Client API library and then invokes the OP-TEE driver of the Linux kernel. The OP-TEE driver performs the secure monitor call, switches the context to the secure world, and reaches the OP-TEE core. In the last stage, the OP-TEE core transfers the request to the target trusted application. After the trusted application completes the request, the system branches back to the calling application. In addition, if the invoked trusted application is not yet loaded into the TEE, the OP-TEE core will make a remote procedure call through a non-secure TEE supplicant to load the trusted application. In this scenario, we send the command for the secret key parameters to the TEE driver and then forward it to the secure side to generate a new symmetric key for decrypting the encrypted image. When all images are decrypted, we use the original function of SWUpdate to install the decrypted images to the eMMC via the dual copy mechanism. If the upgrade process fails, we can reverse the operation by restoring the original partition or logical block.

2.  For the OP-TEE core to call a non-secure remote service, the OP-TEE core invokes the Linux kernel OP-TEE driver and forwards the request to the TEE supplicant daemon, and the system returns the request status to the OP-TEE core.

3.  For a trusted application to invoke an OP-TEE core service, most services must be executed in the privileged mode of the OP-TEE core. The trusted application issues a system call to invoke the corresponding service from the TEE internal core API.

### 3.5. OpenSTLinux Distribution

While functioning as a mainlined open-source Linux distribution, OpenSTLinux distribution is also a key element of the STM32 Embedded Software solution for STM32 multi-market multi-core microprocessors (MPU) embedding a single or dual Arm® Cortex®-A7 core. OpenSTLinux Distribution contains the required packages listed below:

1.  Linux board support package (BSP);
2.  Linux kernel;
3.  Secure boot chain based on ARM trusted firmware (TF-A) and universal bootloader (U-Boot);
4.  Secure OS, open portable trusted execution environment (OP-TEE).

### 3.6. Implementation

We deployed the SWUpdate client application on a customized Linux Kernel and developed a stand-alone trusted application above the customized OP-TEE OS.

We isolated the cryptographic operations as a service to the trusted application. We spent more time determining which part has the most significant impact on security during the progress of developing SWUpdate. Finally, we found the relevant function symbols, install_raw_image and install_from_file included. Most of them will be redirected to the functions that write the image to each block of storage space. Here, we obtained the critical point of writing binary image files to storage and identified whether the image is encrypted and whether it has a SHA256 hash value to compare the decrypted image.

In a nutshell, our proposed update strategy begins with the SWUpdate client, as shown in Figure 9. Then, when the update event arrives, it triggers the SWUpdate daemon to perform further initialization:

1.  The client takes the package from a local external interface or remote server and parses the package description.
2.  The client-side searches for the encrypted image inside the package and initializes TA to obtain the Keyfile and decrypt it. During initialization, the random seed is required to regenerate the parameter Initialization vector (IVT). As Figure 10 shows, the symmetric key generator obtains the new IVT to generate a new AES key.
3.  The next process is to use the AES key to decrypt the packaged image binary file.

4.  It will check whether the hash value is valid.

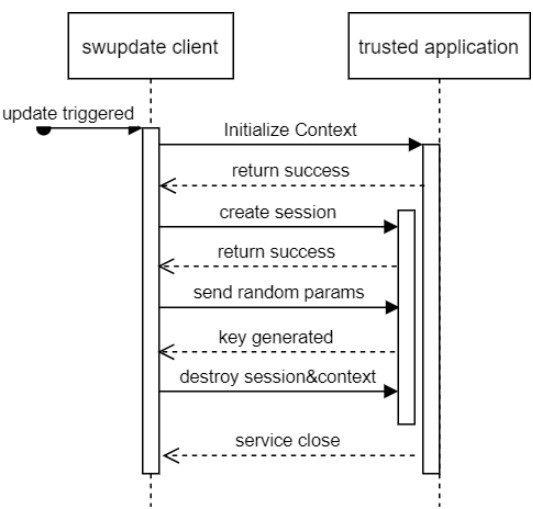

**Figure 9.** The calling convention between the SWUpdate client and trusted application.

```
[Debug] uuid : 0x5dbac793
TEEC_InitializeContext success
D/TA:   TA_CreateEntryPoint:29 === SWUPDATE TA Entry point ===
D/TA:   TA_CreateEntryPoint:30 TA instance created
D/TA:   TA_OpenSessionEntryPoint:45 === SWUPDATE TA Session created ===
D/TA:   TA_OpenSessionEntryPoint:46 session between normal world and secure world
TEEC_Opensession success
Invoking TA to setup the secure key
D/TA:   get_aes_key:118 GET AES KEY : key size 64
D/TA:   get_aes_key:119 GET AES KEY : key return 69efa97810a2ec92a3e23e22a861c92a9ce65074942169a1640d35a6a6a47912
```

**Figure 10.** New key generated by trusted application and used to decrypt the image.

If valid, it will write directly to the defined partition; if not, it will reject the upgrade event and abort the entire process.

## 4. Experimental Result

### 4.1. Demonstrates and Monitors the Transmission between the Server and Client

This section demonstrates our experiments with the STBEAT solution in several steps based on STM32MP157C-DK2 as shown in Table 1. We use the STM32 board for two reasons. First, the CPU family is ARM-based dual Cortex A7, which is the ARMv7 architecture and TrustZone security extensions supported. Second, STM32 has a comprehensive board support package to build and construct applications. The STM32MP157 board also has true random number generators (TRNG), hardware cryptographics, and hash processors. With the support of these functions, it is believed that one can generate a random seed from the TRNG and use the symmetric key to conduct operations based on hardware-based encryption engines. The last component is RAM, in which we need to allocate memory space for key generation and image decryption. By default, the OP-TEE operating system has limited memory for each process so we recompiled OP-TEE to adjust the memory space to utilize more RAM resources.

**Table 1.** Hardware specification.

| | |
|---|---|
| Platform | **STM32MP157C-DK2** |
| Processor | **ARM Dual Cortex-A7 + Cortex-M4** |
| RAM | **4 Gbit DDR3L (16 bits, 533 MHz)** |
| Storage | **Flash, eMMC/SDC (16 GB)** |
| Normal world OS | **OpenST-Linux** |
| Secure world OS | **OP-TEE OS** |
| Symmetric key length | **256 bits** |

The ARM TrustZone is an isolation mechanism to separate an ARM CPU into two logical partitions. Since the original SWUpdate client runs in the normal world, we chose ARM TrustZone as our security solution to deploy. We designed the custom command to be event triggered and to follow the invocation conventions in the GP specification. Moreover, SWUpdate has many supported extension packages, including (a) compressed images using Zlib library and structured language to describe the image using the libconfig library, and (b) support for setting U-Boot variables, GRUB environments, and EFI Boot Guard to give flexibility to the boot process. To sum up, SWUpdate has the feature to update software remotely in an OTA fashion. Our experiment set up the Hawkbit server, an open-source project, and a back-end solution for rolling out software updates to constrained edge devices. After setting up the server, we deploy the update image to the STM32 device as shown in Figure 11, and then SWUpdate will start the upgrade process.

```
GET /default/controller/v1/DEV001 HTTP/1.1
Host: 10.0.2.15:8080
User-Agent: libcurl-agent/1.0
Content-Type: application/json
Accept: application/json
charsets: utf-8

HTTP/1.1 200 OK
Date: Fri, 19 Mar 2021 13:09:37 GMT
Content-Type: application/json;charset=utf-8
ETag: "07dacc6bf54ffc581d544e33974d2fbb7"
X-Content-Type-Options: nosniff
X-XSS-Protection: 1; mode=block
Cache-Control: no-cache, no-store, max-age=0, must-revalidate
Pragma: no-cache
Expires: 0
X-Frame-Options: DENY
Content-Length: 248

{"config":{"polling":{"sleep":"00:05:00"}},"_links":{"deploymentBase":{"href":"http://
10.0.2.15:8080/default/controller/v1/DEV001/deploymentBase/3?c=413476712"},"configData":
{"href":"http://10.0.2.15:8080/default/controller/v1/DEV001/configData"}}}GET /default/
controller/v1/DEV001/deploymentBase/3?c=413476712 HTTP/1.1
Host: 10.0.2.15:8080
User-Agent: libcurl-agent/1.0
Content-Type: application/json
Accept: application/json
charsets: utf-8

HTTP/1.1 200 OK
Date: Fri, 19 Mar 2021 13:09:37 GMT
Content-Type: application/json;charset=utf-8
ETag: "07784a35c0cc67880bc3b40183c9219f3"
X-Content-Type-Options: nosniff
X-XSS-Protection: 1; mode=block
Cache-Control: no-cache, no-store, max-age=0, must-revalidate
Pragma: no-cache
Expires: 0
X-Frame-Options: DENY
Content-Length: 612

{"id":"3","deployment":{"download":"forced","update":"forced","chunks":
[{"part":"os","version":"2.0","name":"ROOTFS","artifacts":
[{"filename":"buildroot.swu","hashes":
{"sha1":"bab586068a42024ef28349b69b1ca360dc876f21","md5":"6a8916925e434349b87df977f5fe2f
d0","sha256":"a6469da3389667349ebc4396281be5df2e1759ae5e8948de97261de4e1616736"},"size":
7845888,"_links":{"download-http":{"href":"http://10.0.2.15:8080/DEFAULT/controller/v1/
DEV001/softwaremodules/1/artifacts/buildroot.swu"},"md5sum-http":{"href":"http://
10.0.2.15:8080/DEFAULT/controller/v1/DEV001/softwaremodules/1/artifacts/
buildroot.swu.MD5SUM"}}}]}]}}POST /default/controller/v1/DEV001/deploymentBase/3/
feedback HTTP/1.1
```

**Figure 11.** The network traffic between Hawkbit server and STM32 board.

### 4.2. Build an Upgrade Package

In the first step, we build a new package with an upgrade image and a description file describing the type of image, including whether it is encrypted and the dynamic IVT value. From this, we need the libconfig library as Figure 12 to parse the description file, and use sysroot from buildroot to make it work.

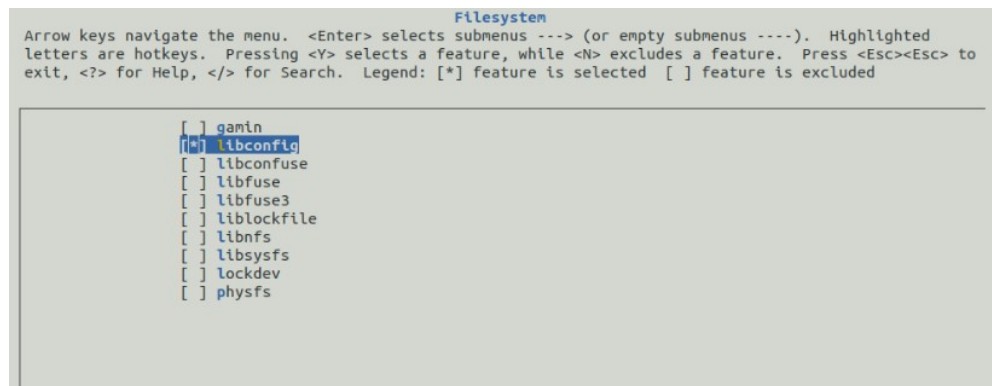

**Figure 12.** Enable libconfig support from menuconfig in buildroot.

*4.3. Comparison*

The upgrade image is encrypted by the symmetric key with the AES algorithm, which is standard and supported by SWUpdate [22]. We use the OpenSSL to generate the AES key and utilize it to encrypt the image. The comparison between the non-encrypted image and encrypted one is presented in Figures 13 and 14.

```
Offset(h) 00 01 02 03 04 05 06 07 08 09 0A 0B 0C 0D 0E 0F   Decoded text

00000000  1F 8B 08 08 14 2D 46 60 02 03 72 6F 6F 74 66 73   .‹...-F`..rootfs
00000010  2E 65 78 74 32 00 EC DD 09 80 55 65 1D 3F FC 73   .ext2.ìÝ.€Ue.?üs
00000020  87 55 40 45 C5 0D 4D 71 CB 1D 07 45 C5 25 C5 05   ‡U@EÅ.MqË..EÅ%Å.
00000030  F7 05 05 97 5C 42 64 93 62 0B 06 77 13 15 73 57   ÷..—\Bd"b..w...sW
00000040  5C D3 C2 A4 B2 B2 B2 B2 D2 52 73 4D 4D CD B5 5C   \ÓÂ¤²²²²ÒRsMMÍµ\
00000050  53 B3 0B 88 12 2E 51 99 5B 2A EF EF B9 F7 8E 73   S³.ˆ..Q™[*ïï¹÷Žs
00000060  18 66 00 1D DF F7 FE FF EF FD 7C EC E9 9C E7 DE   .f..ß÷þÿïý|ìéœçÞ
00000070  73 CF 73 B6 EF 39 CF 39 33 73 C9 32 A0 66 9D 18   sÏs¶ï9Ï93sÉ2 f..
00000080  A5 58 C8 B2 D5 B2 EC A3 F3 0B D9 9E C7 67 59 21   ¥XÈ²Õ²ì£ó.ÙžÇgY!
00000090  FF 7E AF 4A E9 5C AE F6 D8 7C 8F 63 B2 6C C1 82   ÿ~¯Jé\®öØ|.c²lÁ,
000000A0  41 FF 2C 94 A6 2B D7 CB 1A 3F D7 35 CA 94 28 3B   Aÿ,"¦+×Ë.?×5Ê"(;
000000B0  46 D9 B5 2E CB BE D6 3E DE 5B E6 99 B9 AB 3E B1   FÙµ.Ë¾Ö>Þ[æ™¹«>±
000000C0  F7 FD DF FF F1 06 67 FE F8 F1 EF 5F F6 FF D2 DA   ÷ýßÿñ.gþøñï_öÿÒÚ
000000D0  14 9A BF D0 39 37 BE F9 97 E7 1C 37 E8 80 FD 7A   .š¿Ð97¾ù—ç.7è€ýz
000000E0  DF 7C D1 D8 91 0F FD E3 A9 EF 16 62 EA 6E 59 D3   ß|ÑØ'.ýã©ï.bênYÓ
000000F0  7A 75 F9 4F 21 6B 9F FF 70 F7 98 E1 3D 9F A6 F9   zuùO!kŸÿp÷˜á=Ÿ¦ù
00000100  FE 4B 5E A0 D0 3E 5E 1C D2 C7 A1 47 75 FD 67 DE   þK^ Ð>^.ÒÇ¡Guýgþ
00000110  37 BA B7 8B 63 71 C5 28 EB 94 0E D4 3D 3B 6E D6   7º·‹cqÅ(ë".Ô=;nÖ
00000120  78 5A 78 E9 CB DF D8 B3 E3 C3 A7 A7 63 75 A5 F4   xZxéËßØ³ãÃ§§cu¥ô
00000130  7E 5D 63 9C 0A 95 F7 53 ED F9 1F 75 88 6A 8F F4   ~]cœ.•÷Síù.uˆj.ô
00000140  7E BB 2C EB B8 76 D3 FB BF 4D E7 95 CE E7 8D ED   ~»,ë¸vÓû¿Mç•Îç.í
00000150  18 D5 95 D3 FB 11 AC 8F D7 4A EF B7 CB 85 73 FA   .Õ•Óû.¬.×Jï·Ë…sú
00000160  3F 3B C5 7B AB A4 F7 3B 94 CE 33 CD DE 3F 78 ED   ?;Å{«¤÷;"Î3ÍÞ?xí
00000170  CE F1 DE AA E9 FD 8E 2D 7D FE AD 57 96 49 A7 AE   Îñ Þªéý Ž-}þ.W-I§®
00000180  F4 7E A7 52 5E 3F 69 7F 8B 1F A4 F7 87 DE D0 25   ô~§R^?i.‹.¤÷‡ÞÐ%
00000190  AA AB A7 F7 3B B7 F4 F9 0B 37 ED 1A EF F5 4C EF   ª«§÷;·ôù.7í.ïõLï
000001A0  2F D3 52 FB 83 26 74 8B F7 D6 48 EF 77 69 E9 F3   /ÓRûƒ&t‹÷ÖHïwiéó
000001B0  7B 9C B2 6C BC B7 66 7A BF 6B 4B 9F DF E7 FA E5   {œ²l¼·fz¿kKŸßçúå
000001C0  E2 BD 2F A4 F7 BB B5 F4 FE 1F 77 5B 3E DE 5B 2B   â½/¤÷»µôþ.w[>Þ[+
000001D0  BD BF 6C 4B EF DF F8 D3 EE F1 DE DA E9 FD E5 5A   ½¿lKïßøÓîñÞÚéýåZ
000001E0  7A 7F D5 BF AF 10 EF F5 4A EF 2F 1F 67 E5 DC F6   z.Õ¿¯.ïõJï/.gåÜö
000001F0  BF FA ED F4 FE 71 F3 1D E5 2C 4E BB B8 B6 B7 8B   ¿úíôþqó.å,N»¸¶·‹
00000200  2B 78 BB 19 51 8A A9 5E B0 51 A0 46 B4 8F FC B7   +x».QŠ©^°Q F´.ü·
00000210  8F FC B7 8F FC B7 2F A6 BA FC 43 AD E8 10 F9 EF   .ü·.ü·/¦ºüC.è.ùï
00000220  10 F9 EF 10 F9 EF 50 4C 75 F9 87 5A D1 31 F2 DF   .ùï.ùïPLuù‡ZÑ1òß
00000230  31 F2 DF 31 F2 DF B1 98 EA F2 0F B5 A2 53 E4 BF   1òß1òß±˜êò.µ¢Sä¿
00000240  53 E4 BF 53 E4 BF 53 31 D5 E5 1F 6A 45 E7 C8 7F   Sä¿Sä¿S1Õå.jEçÈ.
```

**Figure 13.** The non-encrypted image file with the file signature GZ identified by HxD. In the comparison chart in Figure 14, we cannot see the signature of the document to be recognized.

```
00000000: 29fe fc60 9859 095f 075e 62a4 c3a8 59de    )..`.Y._.^b...Y.
00000010: e23f 0f0a cfbb 8b99 09f8 a9cf 2332 18df    .?..........#2..
00000020: ee74 2529 0dc1 7814 7cb4 6da8 3ace 51af    .t%)..x.|.m.:.Q.
00000030: 63d1 3f1a 4693 39de b851 deaa c466 7f03    c.?.F.9..Q...f..
00000040: 9148 bd93 6bbf 507e 1ac2 29b6 76a8 f7be    .H..k.P~..).v...
00000050: 532c 0d68 4f3e 8bc4 2f3b 6c65 1f66 8286    S,.hO>../;le.f..
00000060: 9321 5e30 61c9 f952 c8fa de9e 0300 e5f6    .!^0a..R........
00000070: d84c 1bf6 cec5 865f 042c 0bfe 883c 3f1e    .L....._.,...<?.
00000080: 3310 5596 47c8 ff19 a3a7 8bec 802c 2405    3.U.G........,$.
00000090: 4213 6bb4 a99a 86ee f131 43df a445 92b5    B.k......1C..E..
000000a0: 1b6d 42e7 fabc a49d 79ea 0433 9048 0053    .mB.....y..3.H.S
000000b0: 29cf 91bb 765a fc89 da57 1260 0712 e014    )...vZ...W.`....
000000c0: a899 e112 2455 9164 8dc1 910f b346 095b    ....$U.d.....F.[
000000d0: f4cd 9234 b099 741a d936 aa43 2ee4 b8dc    ...4..t..6.C....
000000e0: 85f7 04bf c5a7 7917 86e9 76a4 a3d3 c96b    ......y...v....k
000000f0: c0ef 0907 f615 84b3 aed3 5af9 4c14 f976    ..........Z.L..v
00000100: 2900 b2f8 4ca1 5883 b907 0e2d 83f0 3a85    )...L.X....-..:.
00000110: c9b8 7706 715f f183 7978 7299 bc96 0ba8    ..w.q_..yxr.....
00000120: 585b e0ae 00e5 de5b b23a 9855 af40 d00e    X[.....[.:.U.@..
00000130: 7611 8127 de50 6463 a91a 936a f320 b688    v..'.Pdc...j. ..
00000140: d80e 8fd9 4814 462e 409d 7065 585f d4ea    ....H.F.@.peX_..
00000150: 5ec9 7bea 98c2 dd05 20c0 1923 1058 b93b    ^.{..... ..#.X.;
00000160: d566 1ac5 3b96 71b2 b850 156d 3a89 3565    .f..;.q..P.m:.5e
00000170: fd46 0285 2e36 f2cb e977 f913 8624 7857    .F...6...w...$xW
00000180: ad80 e7a4 d67d bc02 f996 ca38 61ad c93a    .....}.....8a..:
00000190: 530c 084a 215f 5228 a66b ef5d 599e b462    S..J!_R(.k.]Y..b
000001a0: 860c 7ac0 a6c0 1bbe d8ae 48aa cde2 0c77    ..z.......H....w
000001b0: 041a e6c8 23ef 82db 9c35 c2d5 0a20 08a6    ....#....5... ..
000001c0: 2a48 56b8 feb4 8a3c 4206 7ad2 3c2c 6fbf    *HV....<B.z.<,o.
000001d0: 6e74 18db 549c c520 4880 5d79 6e5c efe3    nt..T.. H.]yn\..
000001e0: 90ba 5563 153f 8739 9d57 b142 289a 77da    ..Uc.?.9.W.B(.w.
000001f0: ed11 8dc2 6fb5 c0bd 6ba8 0d05 b4c5 fa5f    ....o...k......_
00000200: 9e4d 53b6 c58b 3c7a b807 d043 08bd 9ce8    .MS...<z...C....
00000210: 4c5a b54a 5676 5753 05e0 ab62 9d6a 1b18    LZ.JVvWS...b.j..
00000220: 4af1 333d 75d0 58d5 e231 8fab 5d61 969d    J.3=u.X..1..]a..
00000230: 1668 2b64 ba0e f329 64a3 9ce4 b8ee 9fe6    .h+d...)d.......
00000240: 690d 27d3 8ef0 1f38 eae7 b665 1a44 d008    i.'....8...e.D..
00000250: cf09 8983 77e4 d99f 4413 b01f 49ed ac18    ....w...D...I...
00000260: 4328 c1cb dcba bc3d 4b7f 0b01 4a0d 0866    C(.....=K...J..f
```

**Figure 14.** The image is encrypted by the AES key and we cannot identify the file signature by hexdump.

*4.4. Trusted Application Steps*

In the second step, we check whether the device is ready and in standby mode, then send this new package to the SWUpdate client, whereby it will start the upgrade process as shown in Figure 15. At this moment, it will run the InitializeContext and bring up the trusted application via tee-supplicant by dynamical loading. After the trusted application has been initialized, the SWUpdate client will send our predefined command "TA_SWUPDATE_GET_AES_KEY" to the trusted application.

When the trusted application receives this command, it will calculate the IVT value based on the random seed of the downloaded package. By the way, since we need to output logs from OP-TEE, we use the onboard ST-LINK/V2-1 debugger: Virtual COM port and debug port connected to the host and connected to the board via Putty software. The image shows that the SWUpdate created a session with the trusted application and exchanged random seeds to generate new key parameters.

Eventually, the trusted application moves the generated binary of the key file to the shared memory so the SWUpdate client can retrieve the data from the shared memory to decrypt the image. Then, SWUpdate begins the upgrade progress and writes directly to the partition we specified.

**Figure 15.** SWUpdate obtains the key from the trusted application and decrypts the image to be installed later.

*4.5. Risk and Improvement*

All in all, the image will be decrypted and installed to a predefined partition. Here, we use the single copy mechanism, which works by directly overwriting the original partition. However, there is a risk of power failure occurrence while updating, and there is also a risk that a direct overwrite could damage the currently running system and may cause the system to hang, as shown in Figure 16. Nevertheless, SWUpdate supports a dual copy mechanism to improve the installation process to make it safe and reliable, and we see this implementation as a future improvement.

**Figure 16.** System hang when updating using single-copy mechanism.

**5. Conclusions**

The STBEAT solution is a comprehensive approach to securely update systems and can be applied to resource-limited embedded systems or slightly higher-performance MPU systems. Our scenario uses ARM-based Cortex A7, which is the ARMv7-A architecture. Therefore, it can also run this integrated and complete software system.

To sum up, our STBEAT solution has the following features: (1) Key protection: We place critical security keys in the secure side, accessible only to the secure world. This feature prevents unauthorized or unwanted requests from the normal world. Further, this feature makes key generation more secure than the original method of SWUpdate. (2) System image protection: We encrypt system images with the supported random seed, including kernel or file system or application binaries. In this way, if an attacker performs traffic sniffing, they will not know the file type and the original image file. More importantly, they will not know the decryption algorithm of the random seeds. This can improve security against certain kinds of rainbow table attacks.

*Evaluation*

Our STBEAT system resolves security issues through a novel software update method. System evaluation is also important because the user or any endpoint experience is the impact vector of this solution.

The measurement process is to download the binary file, set the decryption key, and install it on the interface after decryption. The difference between the normal world and the secure world is the TrustZone function. The client sends the command to the trusted application, which receives the command and processes the request. The time spent here is very costly because we have performed a lot of calculations here, including starting TA or generating new decryption keys or decoding parameters for use by the obfuscator. As shown in Figure 17, the average value of the solution with TrustZone disabled was 0.000046 s, and the average value of the solution with TrustZone enabled SWUpdate trusted applications was 0.27 s.

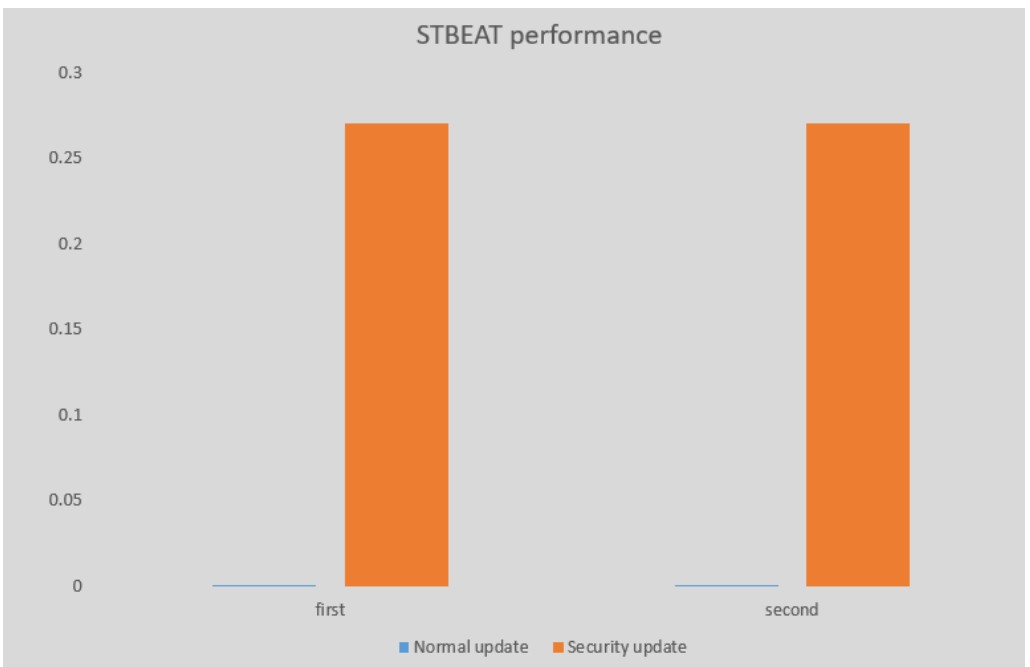

**Figure 17.** The evaluation of STBEAT performance with normal update and secure update.

The increase is as high as 5000%, but we think this situation is acceptable because updates are not frequent in the real world, and the total time spent on user experience is not long.

## 6. Recommendations for Future Work

We can further improve security by generating more random seeds to prevent the cryptographic algorithm from being broken and using more sophisticated algorithms to protect the symmetric key, such as RSA. In addition, we can implement installation checking for storage security. This will verify the address written to the memory is valid and the range belongs to the secure world with the support of the TrustZone protection controller [23].

This paper only implements the integration based on ARM TrustZone and the open-source project SWUpdate. However, we can make improvements as described above to make the STBEAT more complete and comprehensive and submit it to the Yocto or Buildroot projects for integration into the embedded ecosystem whenever possible. Further, we know there are significant architectural differences between ARMv7 and ARMv8. The ARMv8 architecture has more improvements and integrations for TF-A and OP- TEE. We can apply these features to ARMv8 in the future to fit the world's coming product lines.

## 6.1. Discussion

Since the trusted application method we proposed dynamically loaded, the binary file is located in the REE file system. This means that binary files may be vulnerable to attackers.

We simply decompile the trusted application signed by the TEE tool chain, and we can extract some relevant information inside the trusted application. As shown in Figure 18, we use the reverse engineering tool IDA Pro to search for secret strings or any key parameters. We can find the secret string stored in the last part of the binary file. To prevent this, we encode the secret string. As shown in Figure 19, we use IDA Pro to search this again; even though we can extract the string, we cannot know the original state. As shown in Figure 20, we first use the STM32MP1 kernel for our entry point of STBEAT and will grab the binary file to be upgraded from the local storage. Then the SWUpdate client initializes the context and creates a session between the normal world and the secure world. The SWUpdate client triggers the trusted application by sending a predefined command, while the trusted application starts to decode the key string and returns to the CA to decrypt the image. When installing the upgrade image, STBEAT uses a single copy mechanism to install the image to eMMC. At this time, due to the risk of single copy, the currently running file system is damaged. This, however, means that the upgrade image has been successfully written to memory.

```
seg000:0000000000007B50 aWupdateTaSessi_0 db 'WUPDATE TA Session closed ===',0
seg000:0000000000007B6E                   db  74h ; t
seg000:0000000000007B6F aAskFinishedAnd db 'ask finished and session closed',0
seg000:0000000000007B8F                   db  31h ; 1
seg000:0000000000007B90 aA98dc0874c398f db 'a98dc0874c398f7d0954b52cc00bd4fef912504f0401dc4177a40d6c2c50d77',0
seg000:0000000000007BD0                   db  35h ; 5
seg000:0000000000007BD1                   db  32h ; 2
seg000:0000000000007BD2                   db  35h ; 5
seg000:0000000000007BD3 a6545145405544 db '6545145405544',0
```

**Figure 18.** Use IDA Pro to reverse the trusted application to find the secret string in the last part of the binary file.

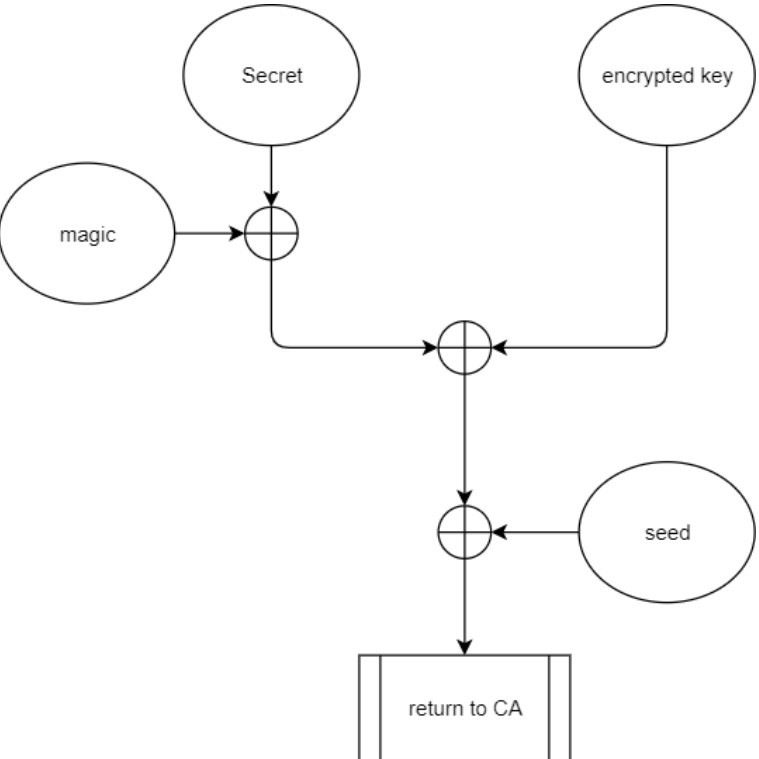

**Figure 19.** Flow chart of magic and secret and encrypted key decoding and seed generation. The secret and magic will perform XOR first, the result will perform XOR with the encrypted key, and then the key is generated by random seed calculation and returned to the CA.

```
root@stm32mp1:~# uname -a
Linux stm32mp1 5.4.56 #1 SMP PREEMPT Wed Aug 5 07:59:52 UTC 2020 armv7l armv7l a
rmv7l GNU/Linux
root@stm32mp1:~# hostname
stm32mp1
root@stm32mp1:~#
root@stm32mp1:~#
root@stm32mp1:~# mount /dev/sda1 /mnt
[  971.124647] FAT-fs (sda1): Volume was not properly unmounted. Some data may b
e corrupt. Please run fsck.
root@stm32mp1:~# cd /mnt
root@stm32mp1:/mnt# swupdate -i /mnt/buildroot.swu -e images,rootfs -K no.key
Swupdate v2020.11.0

Licensed under GPLv2. See source distribution for detailed copyright notices.

[Debug] uuid : 0x5dbac793
TEEC_InitializeContext success
D/TA:   TA_CreateEntryPoint:12 === SWUPDATE TA Entry point ===
D/TA:   TA_CreateEntryPoint:13 TA instance created
D/TA:   TA_OpenSessionEntryPoint:28 === SWUPDATE TA Session created ===
D/TA:   TA_OpenSessionEntryPoint:29 session between normal world and secure world
TEEC_Opensession success
Invoking TA to setup the secure key
[Debug] key : 69ED/TA:  FTA_CloseSessionEntryPoint:40 === SWUPDATE TA Session cl
osed ===
A97810A2D/TEC92AA:  TA_CloseSessionEntryPoint:41 task finished and session close
d
D/T3E22A861C92A9CE65074942169A1640D35A6A6A47912
  [DeA:  TA_DestroyEntryPoint:20 === SWUPDATE TA Destructor ===
bug] ivtD :/ 0A0TA:  TA_DestroyEntryPoint:21 TA instance destroyed
15046CDB5F105EB3C401E58F961D5
[INFO ] : SWUPDATE started :  Software Update started !
Now getting status
[INFO ] : SWUPDATE running :  Installation in progress
```

**Figure 20.** STBEAT starts the upgrade process under the STM32MP1 kernel version.

We force the system to restart, and the machine will check the status of the entire device when it starts, because the newly upgraded operating system does not know this information. After starting, we can see that the buildroot shell has started as shown in Figure 21, and we print out the file system information of the operating system that is built by the buildroot project.

```
Starting klogd: OK
Running sysctl: OK
Initializing random number generator: OK
Saving random seed: OK
Starting network: OK
/etc/init.d/rcS: line 23: /etc/init.d/S98swupdate: Permission denied

Welcome to Buildroot
buildroot login: [    6.663445]  sda: sda1
[    6.668857] sd 0:0:0:0: [sda] Attached SCSI removable disk

Welcome to Buildroot
buildroot login: root
#
#
# uname -a
Linux buildroot 5.4.56 #1 SMP PREEMPT Wed Aug 5 07:59:52 UTC 2020 armv7l GNU/Lin
ux
# [   34.405244] vref: supplied by vdd
[   34.407259] usb33: supplied by vdd_usb
[   34.410941] vref: disabling
[   34.413622] vdda: disabling

#
# hostname
buildroot
# 
```

**Figure 21.** Kernel upgrade successfully and start buildroot shell.

*6.2. Security Analysis*

To conclude, we analyze the important defensive aspects for STBEAT framework:

Firstly, for the critical point, we have to protect the upgraded image, which was modified. With the support of SWUpdate, we encrypt our software image by utilizing the single key algorithm and implementing in trusted application to ensure that the image file

encrypted, as shown in Figure 14, which means that there is no leakage of any relevant information in non-secure world.

Secondly, the trusted applications contain many pieces of critical information based on our development board, and do not have secure storage hardware. Therefore, we use the dynamic loading approach to execute our trusted applications at runtime, but the trusted applications still present some risks in the Linux file system. In other words, the potential problem remains that the trusted application binaries are in a path that anyone can access. We also reverse the engineering to these binaries from the attacker's perspective. We find that we can extract some secret strings from them; see Figure 18.

Since the leakage of the binary string could be harmful to the system, we try to generate another set of unrecognizable strings by obfuscating the secret. The results are in line with our requirements, although it is still necessary to convert the obfuscated string back to the original secret in the trusted application and cost some overhead. Figure 19 expresses the flow chart.

Lastly, due to the current methods and hardware limitations, we use a single copy mechanism, which causes the file system to be corrupted after the entire system is updated and completed, as shown in Figure 16. This even leads to system crash. The availability of the CIA model from the security aspect might be invalidated. In the future, we will utilize the dual-copy method for the system update, and invoke the system image with the other partition to boot after a reboot.

**Author Contributions:** Methodology, Q.-X.H., M.-Y.C. and C.-S.Y. ; software, Huang, Q.-X.H., M.-Y.C. and C.-S.Y.; writing—review and editing, H.-M.S. All authors have read and agreed to the published version of the manuscript.

**Funding:** This research received no external funding.

**Institutional Review Board Statement:** Not applicable.

**Informed Consent Statement:** Not applicable.

**Data Availability Statement:** Not applicable.

**Acknowledgments:** This research was supported in part by the Ministry of Science and Technology, Taiwan, under Project MOST 110-2221-E-007-040-MY3 and MOST 111-2221-E-007-078-MY3.

**Conflicts of Interest:** The authors declare no conflict of interest.

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
