# Peer review of "STBEAT: Software Update on Trusted Environment Based on ARM TrustZone"

_sustainability, doi:10.3390/su142013660_

Round 1

Reviewer 1 Report

The authors proposed a mechanism named STBEAT, aiming to improve safety performance in the process of upgrading the IoT system and updating system files, which is an integration of ARM TrustZone and an open-source project. There are some confusions need to be figured out and some shortcomings to be corrected.

1.     It is noticed that the number of references is 11, which is far from adequate for a carefully researched article, it is better to add more literature with reference value. Some related work and mentioned relevant technologies are without attribution, such as SWUpdate, OpenSTLinux, et cetera.

2.     This article mentioned some challenges need to be solved about IoT devices in the part of abstract, however, discussed work is mainly about security of software update on trusted environment. If proposed mechanism can solve these challenges, there should add more detailed description to explain how does the work figure out them, otherwise it seems to be obscure.

3.     Improve clarity and quality of the figures and ensure uniform style, especially screenshots, which are hard to tell.

4.     It’s better to replenish brief introduction about normal world of TrustZone that the authors only described secure world.

5.     Lack of security analysis and the evaluation about proposed work is inadequate. Consider other evaluation work from different aspects and reorganize it as a separate chapter.

6.     In addition, there are some tiny mistakes in this article. For example, “OPTEE” in abstract should be “OP-TEE”; the first letter of “As” should be lowercase in last paragraph in page 4. Please correct them and check full text carefully to ensure readability and conscientiousness.

Author Response

The authors thank the editor and reviewers for their valuable comments.

In this version of manuscript, the main modifications include the following Response to reviewer in detail. Below we respond to reviewers.

Questions:

The authors proposed a mechanism named STBEAT, aiming to improve safety performance in the process of upgrading the IoT system and updating system files, which is an integration of ARM TrustZone and an open-source project. There are some confusions need to be figured out and some shortcomings to be corrected.

Response to reviewer

  1. It is noticed that the number of references is 11, which is far from adequate for a carefully researched article, it is better to add more literature with reference value. Some related work and mentioned relevant technologies are without attribution, such as SWUpdate, OpenSTLinux, et cetera.

 Thanks for this comments. We add more some relevant articles with related work, and updated the recently journal for hardware security of this issue for SWUpdate, OpenSTLinux.

  1. This article mentioned some challenges need to be solved about IoT devices in the part of abstract, however, discussed work is mainly about security of software update on trusted environment. If proposed mechanism can solve these challenges, there should add more detailed description to explain how does the work figure out them, otherwise it seems to be obscure.

Thanks for this comments. We add more some relevant articles with IoT devices descriptions and recently RISC-V research compared to ours. Figure.9 to 16 shows our implementation process and inspect results in detail.

  1. Improve clarity and quality of the figures and ensure uniform style, especially screenshots, which are hard to tell.

 Thanks for your comments. We improve our figure resolution and describe each figure in the article in section 4.

  1. It’s better to replenish brief introduction about normal world of TrustZone that the authors only described secure world.

 Thanks for your comments. We almost rewrite the TrustZone description from abstract to introduction.

  1. Lack of security analysis and the evaluation about proposed work is inadequate. Consider other evaluation work from different aspects and reorganize it as a separate chapter.

Thanks for your comments, we add some each of steps in Figure 9, and have more evaluation work from our aspects.

  1. In addition, there are some tiny mistakes in this article. For example, “OPTEE” in abstract should be “OP-TEE”; the first letter of “As” should be lowercase in last paragraph in page 4. Please correct them and check full text carefully to ensure readability and conscientiousness.

Thanks for carefully check our text sentences, we double review each sentence, and modify above described mistakes.

Reviewer 2 Report

Title 

STBEAT: Software Update on Trusted Environment based on ARM TrustZone

abstract

- if it is all about IoT, no need to mention automobile. automobile is not mentioned in the content.

- please add brief result and analysis in abstract

- what does ARM stand for?

references

- lask of peer reviewed paper or journal, especially the latest one (published on 2022).

- the research problem must be from peer reviewed articles.

the research design, questions, hypotheses and methods.

- due to the lack of peer reviewed articles, the question and methods are not scientifically and strongly justified.

results

- how good is the result is not properly compared with certain benchmark.

content:

- due to the lack of peer reviewed articles, the content is not succinctly described and contextualized with respect to previous and present theoretical background and empirical research.

- the contribution is there and the related work is also there, however the contribution with regard to the related work is missing or not clear.

Author Response

The authors thank the editor and reviewers for their valuable comments.

In this version of manuscript, the main modifications include the following Response to reviewer in detail. Below we respond to reviewers.

Questions:

abstract

- if it is all about IoT, no need to mention automobile. automobile is not mentioned in the content.

- please add brief result and analysis in abstract

- what does ARM stand for?  

references

- lack of peer reviewed paper or journal, especially the latest one (published on 2022).

- the research problem must be from peer reviewed articles.

the research design, questions, hypotheses and methods.

- due to the lack of peer reviewed articles, the question and methods are not scientifically and strongly justified.

results

- how good is the result is not properly compared with certain benchmark.

content:

- due to the lack of peer reviewed articles, the content is not succinctly described and contextualized with respect to previous and present theoretical background and empirical research.

- the contribution is there and the related work is also there, however the contribution with regard to the related work is missing or not clear.

Response to reviewer

For Abstract part, we rewrite the brief result and analysis, and what does ARM stand for? The answer is that ARM (stylised in lowercase as arm, formerly an acronym for Advanced RISC Machines and originally Acorn RISC Machine) is a family of reduced instruction set computer (RISC) instruction set architectures for computer processors, configured for various environments. The reference link here: https://en.wikipedia.org/wiki/ARM_architecture_family

For references part, the hardware security issue especially in STM32 is fewer on IEEE Xplore, but we try to add relevant journal and conference in recently research from 2019 – 2021 RISC-V issue for compared to ARM architecture. Seeing ref.11-ref.13, please.

For Results part, It’s hard to have more comparison with other article, because these articles are also not clearly compared to our methodology, so we proposed dynamically loaded, the binary file is located in the REE file system. This means that binary files may be vulnerable to attackers.

And others, we try to rewrite to make contents more clear, thanks giving some valuable comments for us.

Reviewer 3 Report

1.       The methodology of the paper is poor. It is not being deliberated anywhere In the manuscript

2.       Literature review has to be improved lot with state of the art surveys.

3.       References are so limited

4.       Authors have not compared any state of the art research attempts

5.       Evaluation measures for validating the accuracy of the proposed approach is lacking

In nutshell, I am not finding suitability of the work for possible consideration in the current form.

Author Response

The authors thank the editor and reviewers for their valuable comments.

In this version of manuscript, the main modifications include the following Response to reviewer in detail. Below we respond to reviewers.

Questions:

  1. The methodology of the paper is poor. It is not being deliberated anywhere In the manuscript

  1. Literature review has to be improved lot with state of the art surveys.

  1. References are so limited

  1. Authors have not compared any state of the art research attempts

  1. Evaluation measures for validating the accuracy of the proposed approach is lacking

In nutshell, I am not finding suitability of the work for possible consideration in the current form.

Response to reviewer

Thanks giving the above comments, this issue is hard to have more references to take from IEEE Xplore website especially in STM32 board, so we try to our best to more analysis in this issue clearly, and also we continue to survey on RISC-V architecture compared to ARM architecture, and we add adequate literatures in references part. Besides there are recently limited journals which cause fewer comparison, but we just can propose the improved methodology compared to previous limited approach.

Reviewer 4 Report

This paper needs to be restructured so that the discussion is after the results section, not after the future work section

In sub Section 1.1 the background needs to be improved and more details added, or this section can be merged with the introduction 

What is the reference for subsection 1.2? 

What is the reference for subsection 1.2? 

Are there only 5 related works in section 2: related work? There are no references in sections 2.1, 2.4, and 2.5. also it would be helpful to add more sources to support your idea. 

References are needed for figures 2 and 3. 

Author Response

The authors thank the editor and reviewers for their valuable comments.

In this version of manuscript, the main modifications include the following Response to reviewer in detail. Below we respond to reviewers.

Questions:

This paper needs to be restructured so that the discussion is after the results section, not after the future work section

In sub Section 1.1 the background needs to be improved and more details added, or this section can be merged with the introduction

What is the reference for subsection 1.2?

Are there only 5 related works in section 2: related work? There are no references in sections 2.1, 2.4, and 2.5. also it would be helpful to add more sources to support your idea.

References are needed for figures 2 and 3.

Response to reviewer

Thanks for the valuable comments, we combine the background with introduction. And the reference for subsection 1.2 is reference 4, so we describe in section contributions. 2.1, 2.4, and 2.5 we add the reference manual source to support our idea. Figures 2 and 3 we drawn by ourselves.

Round 2

Reviewer 1 Report

Some changes and modifications have been seen in the revised version of your paper according to the given comments, however, not comprehensive.

It is better to add security analysis.

Author Response

Thanks for this comments. We add 6.2 subsection to explain comprehensive conception help you understand with Figures and flowchart. 

Reviewer 3 Report

Pl.add more references. The cited references are not sufficient

Also strengthen the literature review section based on the newly added references

Organization of the manuscript can be improved

Author Response

Thanks giving the above comments, we already added some related references from IEEE Xplore website especially in STM32 MCU board, ARM Cortex-A series TrustZone, OP-TEE, SWupdate, and free and open source software(FOSS) articles from transactions journals, and internet of things journals for this issue from ref20 to ref26 clearly, and also we continue to survey on RISC-V, MIPS, x86 architecture compared to ARM architecture. We added adequate literatures in references part. But there are recently limited journals which cause fewer comparison to our methodology, we try to explain our research clearly.

Round 3

Reviewer 3 Report

The newly introduced communication flow diagram is convincing

The architecture diagram is also good.

Experimental evaluation  is also deliberated

Author Response

Response to reviewer:

Thanks reviewer give us such a valuable comments and recognize our research results. Finally, we also do some adjustment of this paper, Figure1, 7, 8 redrew by ourselves, added 6.2 subsection for deeply description of security analysis, and change the references 10, 11 website link to prove our research achievements.